# Quantifying Land Use/Land Cover and Landscape Pattern Changes and Impacts on Ecosystem Services

**DOI:** 10.3390/ijerph17010126

**Published:** 2019-12-23

**Authors:** Qingjian Zhao, Zuomin Wen, Shulin Chen, Sheng Ding, Minxin Zhang

**Affiliations:** College of Economics and Management, Nanjing Forestry University, Nanjing 210037, China; chenshulin0923@163.com (S.C.); dsllt@sina.com (S.D.); zhangminxin@njfu.edu.cn (M.Z.)

**Keywords:** land use, land cover, landscape pattern, index, ESV

## Abstract

Based on satellite remote sensing image, GIS and Fragstats, this study modeled and calculated the dynamic changes of land use, land cover and landscape patterns in Guizhou Province, China, and calculated the changes of ecosystem service values (ESVs). The impacts of the evolution of landscape patterns on the ESVs were analyzed, and reasonable policy recommendations were made. The findings are as follows: (1) In the past two decades, the area of cropland and grassland has decreased; the area of water bodies, urban and rural, industrial and mining, and residential areas has increased; the area of forestland has increased first and then decreased. (2) The two major types of landscapes, cropland and grassland, are clearly being replaced by two land types, forest land and water bodies. (3) Overall, the degree of landscape aggregation and adjacency has decreased, and the landscape heterogeneity has increased. (4) The total amount of ESV in 2000, 2008, 2013 and 2017 was 2574 × 10^8^ Yuan RMB, 2605 × 10^8^ Yuan RMB, 2618 × 10^8^ Yuan RMB and 2612 × 10^8^ Yuan RMB, respectively. The changes of landscape patterns had important impacts on the ESVs. In order to solve the problems caused by the increasingly prominent changes in the landscape patterns and improve the ESVs, it is necessary to rationally plan and allocate land resources, optimize the industrial structures, and develop effective regulatory policies.

## 1. Introduction

Land use is the long-term or regular management of land by humans based on its characteristics. Changes in land use can directly reflect the impact of humans on land resources [1]. In 1995, the International Human Dimensions Programme (IHDP) and the International Geosphere-Biosphere Programme (IGBP) jointly released a scientific research plan for land use and land cover change, opening up a new field of research [2]. In 1999, IHDP and IGBP collaborated again and proposed the “LUCC Change Implementation Strategy” [3]. The nature of land cover in different regions of the world is mainly determined by natural factors. However, due to the intensification of human resource utilization and remediation activities, land cover changes have been greatly affected [4,5,6]. Human production activities such as agriculture, forestry, animal husbandry and urban development are attributed to the land use category, while cropland, forestland, grassland, roads, soil, ice and water bodies fall into the category of land cover.

Land use and land cover changes play a key role in ecosystem service supply and other landscape functions [7,8,9], which is a multidimensional process with multiple driving forces [10]. Maintaining ecosystem functions and ecosystem service flows is a major challenge under the premise of ensuring the sustainable management of forest resources, food security and other socio-economic development needs [11,12,13], as it is likely to generate a strong conflict between land resource supplies and utilization [5,14]. The optimization of land use and land cover depends on the inherent laws, spatiotemporal changes of biophysical conditions and ecosystem structures and functions [15,16,17].

The landscape pattern is the product of long-term interactions between various biotic and abiotic processes in a certain area. It consists of the interactions and arrangements of various landscape elements in the spatial scope [18,19,20]. The landscape pattern mainly shows the spatial structure relationships between the elements, which can be divided into five types of spatial structure: patch, corridor, matrix, network, and edge. Pattern changes are highly interoperable. The change of landscape pattern usually refers to the change of various patches or landscape size, shape and aggregation degree in space, which is the specificity of spatial heterogeneity [21,22,23]. Landscape pattern analysis emerged globally in the 1980s [24]. Shortly thereafter, the study of landscape patterns began to shift from qualitative research to quantitative research [25]. According to the findings of the study, the landscape pattern has a potential impact on the classification and evolution of vegetation [26]. Effectively identifying the relationship between landscape patterns, scale effects, and ecological processes can help to accurately select landscape pattern indices [27]. Regional differences have an indispensable influence on landscape diversity [28]. There is a close relationship between landscape spatial structure, forest cover [29] and urban greening [30]. In recent years, the landscape pattern research became more comprehensive, and was combined with urban development and ecological environment. Rural and urbanization development has a significant impact on landscape pattern changes [31]. Ecological process modeling and land degradation simulation are two kinds of powerful tools for exploring the evolution of landscape patterns [32]. Changes in landscape patterns were found to have strong environmental externalities [33].

As quantitative indicators for analyzing the characteristics of landscape patterns, the landscape indices have developed rapidly in the past 20 years [34,35]. The combination of satellite remote sensing (RS), geographic information system (GIS) and landscape ecology theory has formed a unique research paradigm for large-scale ecosystem spatial patterns [36]. Many landscape pattern models, such as landscape fragmentation models, have been established to study changes in landscape patterns The landscape pattern indices can be generally divided into three levels: patch level, land cover type level and landscape level, which represent the pattern changes of a single patch, land cover type and landscape area, respectively. In general, in the analysis of landscape pattern changes in a large region, the calculation results of patch-level landscape indices do not contribute much to overall landscape pattern changes, so landscape pattern indices are often established from land cover type and landscape levels. With the development of technology, the use of GIS platform and Fragstats software can calculate dozens of landscape pattern indexes [37]. Regarding the impact of landscape on ecosystem services, current research mostly focuses on changes in land use and landscape pattern [38], the ability of landscapes to provide ecosystem services [39], and ecological security and ecological risks [40,41]. There are few articles on the combination of changes in landscape pattern and changes in the ESV of ecosystem services.

This study takes Guizhou Province, China, as the research object. Located in the southwest of China, Guizhou Province has a unique kind of karst landform, which is ecologically fragile. The environment situations of soil erosion and rocky desertification are extremely serious [42,43], and the landscape pattern changes have great negative impacts on ecosystem service supply [44,45]. This study attempts to reveal the spatial-temporal changes in land use and landscape patterns, as well as the impact on ESV, to provide a reasonable scientific basis for the management and use of land resources.

## 2. Materials and Methods 

### 2.1. Study Area 

Guizhou Province is located at 24°37′–29°13′ north latitude and 103°36′–109°35′ east longitude. It is a transportation hub in southwest China and an important part of the Yangtze River Economic Belt. Guizhou Province has a total area of 176,167 square kilometers, accounting for 1.8% of the country’s total area. And it is geographically located in the eastern part of the Yunnan-Guizhou Plateau. The map of Guizhou Province is shown in Figure 1. It is inclined from the central to the north, east and south, with an average elevation of approximately 1100 meters. The types of landforms in Guizhou have diverse characteristics, including plateaus, mountain plains, mountains, hills, platforms, basins, and river terraces. These different types of landforms not only have different forms and altitudes, but also have different causes and constituent materials. Plateaus, mountains, and mountains account for approximately 87% of the province’s total area, hills for 10%, and basins, river terraces, and valley plains for only 3%. The types of karst landforms in Guizhou are complete and widely distributed. The exposed area of carbonate rocks accounts for 73% of the total area of the province. The ecological environment in Guizhou Province is very fragile, and soil erosion and rocky desertification are extremely serious [46]. In 2018, the area of soil erosion control in Guizhou Province was 683.5 km^2^. The imperfect government system and the lack of public awareness of environmental protection, human factors such as transitional grazing, deforestation and socio-economic activities have further exacerbated the local ecological and environmental situation. According to the Guizhou Province Statistical Yearbook, as of 2018, Guizhou’s resident population was 36 million. In recent years, the economy has developed rapidly, and the GDP growth rate for the third consecutive year is among the top three in the country. However, compared with other provinces, the infrastructure construction is still relatively backward, and the natural geographical environment has become a major factor restricting the economic development of Guizhou Province. Reasonable planning and protective development of land resources is one of the important factors to promote rapid economic development. The basic information for this study came from land use and land cover data in Guizhou Province, which were interpreted from satellite remote sensing images from 2000, 2008, 2013 and 2017, as well as economic and social development data. The land use data is mainly used for image processing and spatial analysis by means of ENVI 5.1 (ITT Visual Information Solutions, USA) and ArcGIS 10.2. (ESRI, RedLands, USA) In total, 12 types of landscape indices were calculated using Fragstats 4.2. SPSS was used to calculate and analyze the impacts of landscape pattern changes on ESVs. The maps of land use classification in different periods are shown in Appendix A.

### 2.2. Methods

(1) Land use changes and transfer. The comprehensive land use dynamic degree and the single land use dynamic degree constitute another popular framework to analyze temporal change among land categories [47]. The land use type dynamic degree, also known as the land use change rate index, is mainly used to calculate the quantitative value of a land use type change, and can also be used to estimate the land use change trend and the change speed in the next few years. Its equation is as follow.
(1)$$K=\frac{U_{b}-U_{a}}{U_{a}}\times \frac{1}{T}\times 100\%,$$
where K indicates the change rate of a land use type during the study period; $U_{a}$ indicates the area of the land use type at the beginning of the study period; $U_{b}$ indicates the area of the land use type at the end of the study period; T indicates the study period.

The land use transfer matrix is a matrix that describes the change in the number of areas and the trend of transfer between various land use types in different periods. The equation is as follows.
(2)$$C_{i\times j}=A_{i\times j}^{k}\times 10^{n}+A_{i\times j}^{k+1},$$

In the equation, n is generally 1 or 2. When the number of land use types, m < 10, n is 1; when the number of land use types, 10 < m < 100, n is 2, etc. $A_{i\times j}^{k}$ and $A_{i\times j}^{k+1}$ refer to the two types of land use type maps, respectively. $C_{i\times j}$ refers to the land use type change matrix from the k period to the k+1 period.

(2) Landscape indices. Studies have shown that there is often a high degree of correlation between many landscape indices and it is easy to confuse them. This requires researchers to be able to select landscape indicators that are consistent with the status of the study area from a variety of indicators [48,49,50]. Based on the characteristics of land use area changes and spatial transfers in Guizhou Province, 12 independent landscape pattern indices that can fully reflect the patch area, shape, concentration and diversity of Guizhou Province were selected as landscape pattern research indicators. The landscape pattern index set includes the Patch Number (NP), Patch Density (PD), mean of patch area (AREA_MN), Largest Patch Index (LPI), Landscape Shape Index (LSI), area-weighted mean (SHAPE_AM), Aggregation Index (AI), Contagion Index (CONTAG), Perimeter–Area Fractal Dimension (PAFRAC), Interspersion and Juxtaposition index (IJI), Shannon Diversity Index (SHDI) and Shannon Evenness Index (SHEI) [51]. The equations for the indices are shown in Appendix A. Using the above indices, the landscape pattern changes are analyzed from the land cover type level and landscape level. Seven kinds of indices were used to analyze changes in the number and area of patches, landscape shape characteristics, and degree of landscape aggregation at the land cover type level. The equation for each index is shown in Appendix A. Nine kinds of indices were used to analyze the landscape pattern changes in landscape area and quantity, landscape shape characteristics, landscape aggregation degree and landscape diversity characteristics at the landscape level. The equation for each index is shown in Appendix A. 

(3) ESV assessment models. Ecosystem services refer to the ability of an ecosystem or ecological environment to directly or indirectly provide raw materials, goods and services required by humans in a certain area [52]. Land use area changes, structural changes, and spatial transfers all have an impact on the number of patches in the landscape, the shape of the patches, and the degree of patch aggregation, which lead to changes in ecological processes such as the function, structure, and value flow of ecosystems, and ultimately affect output of ecosystem service functions. The supply of ecosystem services, regulation services, support services, and cultural services also fluctuates. The impact of changes in land use and landscape patterns on ecosystem services is mainly reflected in changes in the value of ecosystem services [53,54]. Robert Constonza first proposed the equivalent method of ecosystem service values for ecosystem value assessment [55]. Based on the geographical and climate situation in Guizhou Province, the ESV Equivalent Table was examined [56,57]. The average biomass factor of the main forest species, *Pinus*, *Pinus armandii* and other types of pines, is 1.25, which corrects the ESV equivalent of forestland [58]. Thus, the ESV per unit area of the terrestrial ecosystem in Guizhou Province was obtained as shown in Table 1. Based on the changes in land use area, structure and spatial distribution, the ESVs can be calculated and analyzed. The ESV models are as follows.
(3)$$ESV=\sum_{k=1}^{n}A_{k}\times VC_{k},$$
(4)$$ESV_{f}=\sum_{k=1}^{n}A_{k}\times VC_{fk},$$
where ESV indicates the total value of ecosystem services. $ESV_{f}$ indicates the value of the f−th item of the ecosystem. $VC_{fk}$ indicates the value of the f−th item of the ecosystem service of land use type k. $A_{k}$ indicates the area of land use type k (ha). $VC_{k}$ indicates the ESV coefficient—that is, the ESV (Yuan RMB·ha^−1^·a^−1^) of land use type k per unit area.

(4) Ecosystem service gain and loss matrix. Ecosystem service gain and loss refers to the loss and gain of ESV caused by the conversion between land use types. The gains and losses of ESV can be calculated based on the conversions of land use types and ESV coefficients. The equation of the ecosystem service gain and loss matrix is as follows.
(5)$$P_{ij}=(VC_{j}-VC_{i})\times A_{ij},$$

$P_{ij}$ indicates the gain or loss caused by the conversion from land use type i to land use type j. $VC_{i}$ and $VC_{j}$ indicates the ESV coefficients of land use type i and land use type j, respectively; $A_{ij}$ indicates the area converted from land use type i into land use type j.

## 3. Results

### 3.1. Changes of Land Use and Land Cover

The land use data from 2000 to 2017 was calculated and compiled. The area of various land types and their changes are shown in Figure 2 and Figure 3. On the whole, in the past two decades, the structure of the six major land types in Guizhou Province has changed significantly, and the proportion of each land type has changed greatly. The proportion of urban and rural areas, industrial and mining, and residential land has changed the most. From 2000 to 2017, the proportions of various land use types in Guizhou Province are ranked as follows: forestland > cropland > grassland > urban and rural, industrial and mining, residential land > water bodies > unutilized land. Of which, the proportions of cropland and grassland decreased year by year; the proportion of forestland increased first and then decreased; the proportion of urban and rural areas, industrial and mining, and residential land and water body area increased year by year; the proportion of unutilized land had not changed much. The proportion of forestland reached more than 50%, exceeding the sum of cropland and grassland; the proportion of cropland decreased from 28.2% to 27.37%. The proportion of forestland increased from 53.10% to 54.19% in 2000–2013, but decreased from 54.19% to 53.92% in 2013–2017. The proportion of grassland decreased the most, from 18.10% to 16.60% during the study period. Urban and rural areas, industrial and mining, and residential land accounted for the largest increase, from 0.34% to 1.47%. The water body area increased year by year, from 0.23% to 0.62%.

Based on the Markov transfer matrix and the spatial overlay analysis function in ArcGIS, the four remote sensing images of 2000, 2008, 2013 and 2017 were superimposed and analyzed to obtain the land use transfer matrix of the study area. The land use transfer matrices for the three periods from 2000 to 2017 are shown in Appendix A. In 2000–2008, the land use type with the largest transferred out area was cropland, with a total area of 348,457 ha, net area of 6514 ha, mainly converted to forestland, urban and rural, industrial and mining, residential land, and grassland; this was followed by grassland, with a total transferred out area of 305,049 ha, net transferred out area of 152,635 ha, mainly converted to forest land, cropland and water bodies. The largest proportion of transferred out area was unutilized land, with a transfer rate of 25%; this was followed by grassland, with a transfer rate of 10%. In 2008–2013, the largest type of land transferred out was cropland, with a total transfer of 331,414 ha and a net transferred area of 50,508 ha, which was mainly transferred to forestland, grassland and urban-rural, industrial, mining, and residential land; this was followed by forestland, with a transferred out area of 275,231 ha, mainly converted to cropland, grassland and urban and rural, industrial and mining, and residential land. The largest proportion of transfers was unutilized land, with a transfer rate of 14%; this was followed by urban and rural, industrial, mining, and residential land, with a ratio of 7%. In 2013–2017, the largest transferred out area was cropland, with a transfer of 369,928 ha and a net transfer of 90,209 ha, mainly to forestland, grassland and urban-rural, industrial, mining, and residential land; this was followed by forestland, with a total transfer of 295,605 ha, net transfer of 46,930 ha, mainly converted to cropland, grassland and urban and rural, industrial and mining, and residential land. The type of land with the largest transfer rate was unutilized land, with a transfer rate of 16%; this was followed by urban and rural, industrial, mining and residential land, with a transfer rate of 9%.

### 3.2. Landscape Pattern Indices at the Land Cover Type Level

The calculated values of the seven kinds of indices at the land cover type level are shown in the Appendix A.

(1) The NP and PD. The 30*30 raster data for 2000, 2008, 2013 and 2017 was imported into Fragstats 4.2 to calculate the landscape indices. The results are shown in Figure 4 and Figure 5. As can be seen from the two figures, during the study period, the NP and PD of each land use type are sorted by size as follows: cropland > forestland > grassland > urban and rural, industrial and mining, and residential land > water bodies > unutilized land. The NP and PD of cropland were the largest, while the NP and PD of unutilized land are the smallest. Overall, the total NP of Guizhou Province increased during the period 2000–2017, which caused an increase in the degree of landscape fragmentation. Cropland contributed a lot to the total NP, accounted for 65%. The NP of cropland decreased by 4819 in 2000 to 2008, and increased by 4421 in 2008 to 2017, indicating that the degree of fragmentation of cropland has eased first then deepening. Heterogeneity is constantly strengthening. According to calculations, the largest land use type is forestland, accounting for more than 50%. The NP and PD of the forestland remained stable throughout the study period. The NP decreased by 1408 in 2000–2008, increased by 1572 in 2008–2013, then decreased by 133 in 2013–2017. Compared with cropland, the degree of fragmentation of forestland is lower, and the spatial heterogeneity is also weaker. Grassland is the third largest land cover type, accounting for 6% of the total area, and its NP and PD fluctuated widely. Compared with forestland, the development and utilization of grassland is greater, and the degree of fragmentation is much higher. Overall, the accelerated urbanization process in Guizhou Province in recent years has led to the transfer of large areas of forestland, grassland and cropland.

(2) The AREA-MN. The calculation results of the index AREA-MN are shown in Figure 6. In 2000, 2008, 2013 and 2017, the AREA-MN of cropland was 67.619, 72.344, 68.680 and 65.973 ha, respectively, with a downward trend in the later stage. The AREA-MN of forestland fluctuated, rose firstly, then decreased, and then rose slightly. In 2000–2008, the AREA-MN increased by 44.495, and then decreased significantly, dropping by 38.738 after 2008, showing a sign of deterioration of fragmentation. However, the AREA-MN is larger and much higher than the other five kinds of landscape types. So, on the whole, the degree of fragmentation of forestland is smaller than that of other land cover types. The AREA-MN of grassland in 2000, 2008, 2013 and 2017 was 185.999, 214.544, 183.668 and 187.355 ha, respectively, rising first, then decreasing, and then rising. The AREA-MN of grassland increased by 28.545 in 2000–2008 and decreased by 30.876 in 2008–2013. During the study period, the total area of grassland decreased, and the fragmentation situation became worsen. The AREA-MN of the water body area increased year by year with an improved degree of fragmentation, which was conducive to the enhancement of the ESV.

(3) The LSI. The LSI indices are shown in Figure 7. According to the LSI, the land cover types are sorted as follows: cropland > forestland > grassland > urban and rural areas, industrial and mining, and residential land > water area > unutilized land. The cropland landscape is the most irregular, and the patch edges were constantly growing. The shape of the water bodies landscape is the most regular and simple. During the study period, the LSI of cropland showed volatilities. The degree of fragmentation of cropland continued to increase, with an increased spatial heterogeneity. The cropland resources were unevenly distributed and gradually evolved from a continuous and single geographical form to a broken and complex landscape. Compared with cropland, the LSI of forestland is small, and the overall fluctuation is volatility. The NP and PD of forestland were stable, indicating that the forestland was the dominant landscape type among the three major land types. Further, the degree of fragmentation was low, and the spatial heterogeneity was weak. However, the shape of the forestland landscape gradually became irregular after 2013. The LSI of grassland showed a downward trend in 2000–2013, and then a smaller upward trend in 2013–2017. The LSI of urban and rural areas, industrial and mining, and residential land has been increasing from 2000 to 2017. It is the landscape type with the largest increase and the highest rate of increase in the LSI among the six kinds of land use. Compared with cropland, grassland and forestland, its degree of fragmentation has been alleviated and improved.

(4) The SHAPE_AM. The SHAPE_AM of each land type area from 2000 to 2017 is shown in Figure 8. The size order is as follows: forestland > cropland > grassland > water body area > urban and rural areas, industrial and mining, and residential land > unutilized land. The SHAPE_AM of forestland was the largest among the six land types. However, its landscape shape was more regular and simpler than cropland and its degree of fragmentation was lower than that of cropland and grassland, which indicates that although the forestland distribution was more continuous, its spatial complexity was the highest, and its landscape shape was the most complex.

(5) The AI. The AI of each land cover type in 2000–2017 is shown in Figure 9. The order of aggregation indices is expressed as forestland > grassland > unutilized land > urban and rural areas, industrial and mining, and residential land > cropland > water bodies. It shows that the patches of forestland have the biggest AI with strong interconnection. However, in 2013–2017, the AI of forestland patches decreased, accompanied by an increase in fragmentation. The AI of grassland fluctuated approximately 94 with stable fluctuations. And the patches were well connected to each other with high AI. Compared with forestland and grassland, the aggregation and connectivity between patches of cropland were poor, and the AI was decreasing year by year.

(6) The IJI. As can be seen from Figure 10, all the IJIs were on the rise. The adjacency between patches has been increasing, but the trend has slowed down, which is the typical characteristic of artificial landscapes. The degree of connection between grassland, cropland, forestland and other patches was at a moderately lower level. The degree of bridging between water bodies, urban and rural, industrial and mining, and residential land was at a medium level. The IJI of forestland and cropland were both low, not exceeding 40, indicating that the ridges of forestland and cropland were poorly connected to the patches of other land cover types. The degree of aggregation between forestland patches was the largest, and the state of continuity distribution and degree of fragmentation were also in good condition. Although the shape of the forestland landscape was the most complex, the degree of connection between forestland patches and other types of patches was always increasing. Cropland had the highest degree of fragmentation, and its patches were the most irregular ones, tending to be more complicated year by year. Cropland had the most severe landscape state.

### 3.3. Landscape Pattern Indices at the Landscape Level

The calculated values of the nine indices at the landscape level are shown in the attached Appendix A. 

(1) The NP and PD. As shown in Figure 11, during the period 2000–2017, the trends of the NP and PD in Guizhou Province were the same, from 2000 to 2008, and from 2008 to 2017. In the previous stage, the overall fragmentation of the landscape pattern gradually improved; however, in the latter stage, the situation reversed, and the NP and PD increased significantly. Figure 12 shows the trend of the LPI of the landscape. The LPI increased by 0.997 from 2000 to 2013, indicating that the largest patch area in the landscape has increased. However, in 2017, the LPI was greatly reduced and the proportion decreased. The interaction between the NP, PD and PI means that the degree of fragmentation of the landscape is increasing under the influence of human activities.

(2) Characteristics of landscape shape. As can be seen from Figure 13, the LSI in 2000, 2008 and 2013 was 387.133, 386.436 and 378.908, respectively. As can be seen from Figure 13, the LSI had a significant downward trend in 2008–2013, indicating that although the landscape shape was irregular, it was moving toward mitigation and improvement. However, during the period of 2013–2017, the irregularity of the landscape pattern increased sharply, the edges of the landscape patches lengthened, and the degree of fragmentation increased. During the whole study period, the perimeter–area fractal dimension (PAFRAC) value was approximately 1.5, showing a volatility change, indicating that human activities have a greater impact on the landscape pattern changes.

(3) Characteristics of landscape pattern aggregation. The landscape pattern characteristics can be analyzed by two indices, the IJI and CONTAG. As can be seen from Figure 14, the CONTAG increased slightly first, then decreased significantly. The value fluctuated within the range of 60–65, indicating that the landscape aggregation and contagion of Guizhou Province were at a medium level. The decrease in the CONTAG indicated that the degree of fragmentation of the land landscape has increased. There were more small patches in the landscape, and the patches were intermittently distributed. The degree of landscape aggregation and contagion was weakened, and the connectivity of the dominant landscape has also declined. The trend of the IJI was opposite to the trend of the CONTAG, with slightly rising after a small decline, fluctuating approximately 40, indicating low bridging between different patch types. 

(4) Characteristics of landscape diversity. The SHDI and SHEI were chosen to represent landscape diversity, as shown in Figure 15. For the landscape system, the richer the patch types and the higher the degree of fragmentation, the greater the amount of uncertain information contained in the landscape, and the weaker the stability of the land ecosystem. The SHDI and SHEI showed the same trend, with a slight decline in 2000–2008 and a significant increase in 2008–2017. It can be seen that during the previous period, there were a low patch diversity and strong dominance, with higher richness and fragmentation degree. However, during the latter period, the land landscape diversity showed an upward trend. The corresponding landscape richness and the degree of fragmentation increased.

### 3.4. ESVs

(1) ESVs. The attached Appendix A shows the calculation results of various types of ecosystem service values in different periods. The ESVs generated by various land cover types and ecosystem functions are shown in Figure 16 and Figure 17, respectively. From Figure 16, we can find that the ESV generated by forestland is much greater than that of other land cover types, followed by cropland and grassland. Figure 17 ranks the ecosystem service functions according to the contribution to ESV as follows: Soil formation and conservation > Gas regulation > Biodiversity conservation > Water conservation > Climate regulation > Raw material > Waste treatment > Entertainment and culture > Food production. Among them, the contribution of soil formation and protection function is the greatest, and the contribution of entertainment culture and food production function is the smallest, indicating that the value of ecosystem culture and supply function is far less than the value of supporting and regulating service functions. The increasing trend of climate regulation and gas regulation ESV was mainly due to the increasing annual forest area, which helped to adjust the value of service. The rising trend of ESV in water conservation was mainly attributed to the increase in water area and the good pattern of water landscape. Among the support services, the value of soil formation and protection functions contributed the most to the total ESV, accounting for approximately 17%. The ESV of soil formation and protection service functions and biodiversity conservation service function all showed an upward trend in 2000–2013 and a downward trend in 2013–2017. In supply services, food production functions and raw material functions contributed little to the total ESV. The entertainment and cultural service functions contributed little to the total ESV but kept increasing year by year.

(2) Gains and losses matrix of ESVs. The changes in ESV caused by the transfer between different land cover types can be regarded as the gain and loss flows. The calculation results are shown in Table 2. The flows of gains and losses generated by all cases of shifting to water bodies were positive. All the gain and loss flows generated by the conversion to urban and rural areas, industrial and mining, residential land and unused land were negative, unfavorable for the total ESV. The losses of ESV were mainly caused by the conversion of forestland to cropland, grassland and urban and rural areas land, and from cropland and grassland to urban and rural areas land. The losses of ESV caused by the conversion of forestland to cropland were the highest, 4452.025 × 10^6^ Yuan RMB, accounting for 64.85%. During the study period, the ESV increased by 3757.592 × 10^6^ Yuan RMB, mainly from the conversion of cropland to forestland and water bodies, grassland to forestland and water bodies. The overall gain was greater than the losses, mainly due to the contribution of forestland and water bodies.

(3) Sensitivity analysis. The purpose of introducing sensitivity is to verify the accuracy of the ESV coefficient of each land cover type and the dependences of the total ESV on the coefficients. The equation for the sensitivity index is as Equation (6).
(6)$$CS=\left|\frac{\frac{ESV_{j}-ESV_{i}}{ESV_{i}}}{\frac{VC_{jk}-VC_{ik}}{VC_{ik}}}\right|,$$
where, CS indicates the sensitivity index. $ESV_{j}$ and $ESV_{i}$ indicate the values of the adjusted and initial ecosystem services, respectively. $VC_{j}$ and $VC_{i}$ are the adjusted and initial ESV factors, respectively. Parameter *k* represents one kind of land cover type.

According to Equation (6), the ESV coefficient of each land use type increased or decreased by 50%, and the sensitivity index changes are shown in Table 3. The sensitivity index for all land cover types is less than one. The sensitivities of cropland, grassland and water bodies are small, with sensitivity coefficients less than 0.1. The sensitivity coefficient of forestland is the highest, 0.878–0.881, which means that when the forest ESV coefficient increases or decreases by 1%, the total ESV increases or decreases by 0.878%–0.881%. This shows that the value coefficient adopted is inelastic and suitable for the actual situation, and the results of ESVs are credible.

## 4. Discussion

### 4.1. Correlations between Landscape Patterns and Ecosystem Services

Land use changes would affect the pattern, structure and distribution of land landscape patches, thereby affecting the functions of ecosystem services in the region, leading to the degradation of some ecosystem service functions and reduced ecological effects. Based on the research results of land use, landscape pattern and ecosystem service value, the SPSS software was used to analyze the correlation between the landscape indices and ecosystem service value to explore the impact of land use landscape pattern changes on ecosystem service. The correlations between the six landscape pattern indices of different land cover types, the NP, LSI, AREA_MN, SHAPE_AM, IJI, AI, and the total ESV were calculated. The results are shown in Table 4. There was a clear correlation between the ESV and SHAPE_AM, AREA_MN and AI. A strong negative correlation was found between the ESV and the IJI. No significant correlation was detected between ESV and the NP or LSI. The correlation coefficient between ESV and SHAPE_AM is 0.999, which is significant at the level of 0.01, indicating a very strong correlation. This shows that the higher the complexity of the landscape pattern, the higher the ESV of the ecosystem, and the complexity of the landscape pattern has a positive role in promoting the value of ecosystem services. The correlation coefficient between ESV and AREA_MN is 0.948, which is significant at the level of 0.01, indicating that the correlation between ESV changes and AREA_MN is extremely high—that is, the degree of fragmentation of the landscape and the change in ESV are strongly correlated. When the value of the AREA_MN index continues to increase, the ESV also increases—that is, the smaller the fragmentation of the landscape, the larger the ESV. The correlation coefficient between ESV and the AI is 0.769, which is significant at the 0.01 level, and the correlation is also strong, indicating that the better the degree of landscape aggregation, the higher the ESV. The correlation coefficient between ESV and the IJI is −0.671, which is significant at the 0.01 level, showing a strong negative correlation, indicating that the higher the value of the IJI, the lower the ESV.

### 4.2. Impacts of Landscape Pattern Changes on Ecosystem Services

(1) Impacts on ecosystem supply services. In addition to food production and raw material supply, ecosystem supply services include the production of aquatic products, vegetables, oilseeds, flue-cured tobacco, fruits and meat. During the period of 2000–20017, with China’s ’Grain-for-Green’ Program development, the area and proportion of cropland decreased significantly. From the perspective of landscape pattern, the degree of fragmentation of cropland, the degree of patch aggregation, the degree of adjacency and the intensity of human disturbance have all been increasing, which has led to a continuous decline in grain output, directly affecting the supply of ecosystem services. From the perspective of the structure of the planting industry, high-efficiency economic crops such as vegetables, fruits and medicinal materials have developed rapidly. The proportion of economic crops has been increasing, making up for the large losses of ESV caused by the decline in food production. The change of landscape pattern of cropland had positive effects on the ESV of supply services and the agricultural structure adjustment. However, this trend had also broken the rational use of cropland and other types of land landscapes, resulting in the continued deterioration and destruction of the cropland landscape pattern, which is not conducive to the long-term balance of the economy and ecosystem.

(2) Impacts on ecosystem regulation services. Ecosystem regulation services mainly include gas regulation, climate regulation, water conservation and waste disposal. Cropland, forestland, urban and rural land, industry, mining, and residential land have significant impacts on ecosystem regulation services. Unreasonable planning of towns, blind expansion of land development, and increasingly industrial pollution all have seriously negative impacts on ecosystem regulation services. From the results, during the period of 2000−2013, the ESV of regulation services and its proportion increased year by year. In fact, it does not seem to be as good as it seems. The ESV changes in cropland, urban and rural areas land were negative, which hindered the ESV regulation services. The new ESV added by forestland offset this loss. During the period of 2013−2017, as the forestland area and its ESV began to decline, the urban and rural areas, industrial and mining areas and residential land area increased significantly, resulting in the losses of total ESV far greater than the gains. At the same time, as the ESV of climate regulation and gas regulation functions decreased, the water body area increased significantly, which led to an increase in ESV for water body conservation and waste disposal, thereby offsetting ESV losses due to gas emissions and climate changes.

(3) Impacts on ecosystem support services. Soil formation and conservation and biodiversity conservation are the main forms of ecosystem support services. The former is closely related to the changes of forestland and cropland landscape patterns, and the latter is closely related to the change of forestland, grassland and water bodies landscape patterns. During the period of 2000−2013, the area of cropland in Guizhou Province decreased fast, and the utilization intensity and fragmentation of cropland reached the highest point. However, the area of forestland continued to increase, and the spatial distribution of landscape pattern was optimized. The area of improvement increased from 2,046,500 to 5,816,500 ha, and soil erosion control was strengthened. The improvement of soil formation and protection has offset the losses of ecosystem support services caused by cropland. During the period of 2013−2017, the function, structure and layout of the landscape pattern changed significantly. The area of cropland and forestland decreased significantly, reaching 90,218 and 46,926 hectares, respectively. The soil fertility decreased, and the erosion of runoff on soil deteriorated. Soil formation and protection functions were severely damaged. The ability to acquire, store, treat and recycle nitrogen, phosphorus and other nutrients was also reduced. During the study period, the number of landscape patches in cropland and forestland continued to increase, the degree of fragmentation increased sharply, the degree of landscape aggregation and adjacency decreased, and the connectivity of dominant landscape, forestland, began to decline. Biological habitats were severely damaged. The ESV of biodiversity decreased by 0.017 × 10 ^10^ Yuan RMB in 2013−2017, which had a negative impact on the total ESV of ecosystem services. 

(4) Impacts on ecosystem cultural services. As an important part of ecosystem services, ecosystem cultural services mainly refer to the leisure, entertainment, art, aesthetics and other spiritual and cultural services of the landscape. During the period of 2013−2017, the number of provincial forest parks in Guizhou increased from 75 to 97, and the number of national forest parks increased from 22 to 30, and the total number of national water landscapes reached 31. Historical and cultural landscapes have also been improved and improved. Each ancient town makes full use of its unique resources to promote the tourism economy and promote the continuous improvement of ecosystem services. The construction of supporting facilities in the city has been increasing, which has improved the functions and values of ecosystem cultural services. Forestland and water bodies are the two most important landscapes of ecosystem cultural services. The increase in area and the optimization of landscape pattern have greatly enhanced the value and function of ecosystem services. The value of cultural services increased by 0.035 × 10 ^10^ Yuan RMB during the period 2000−2017.

## 5. Conclusions

The current studies on ecosystem service value mostly focus on value evaluation, value-driven analysis, and the calculation of single ecosystem service value. Few comprehensive studies have combined landscape pattern changes and ecosystem service values. In large-scale areas, studying the correlation and interaction mechanisms between changes in landscape patterns and socio-economic development can better highlight the interaction mechanisms between landscape patterns and human activities. This research modeled and calculated the dynamic changes of land use, land cover and landscape patterns in Guizhou Province, China, and analyzed the responses of ESVs. The characteristics of land use and landscape pattern changes are as follows: (1) During the period of 2000−2017, the area of cropland and grassland always decreased; the area of water bodies and urban and rural areas, industrial and mining, and residential areas increased; the area of forestland increased first and then decreased. (2) The dominant landscape types, cropland and grassland, are gradually being replaced by two types of land types, forestland and water bodies. Forestland is the most dominant landscape type, which has a low degree of fragmentation and a regular landscape shape. The water landscape is in good condition, but the degree of fragmentation has an increasing trend, which requires reasonable protection and utilization. (3) Overall, the degree of landscape aggregation and adjacency has decreased, and the landscape heterogeneity has increased.

The characteristics of ESV are as follows: (1) Total amount of ESV in 2000, 2008, 2013 and 2017 was 2574 × 10^8^ Yuan RMB, 2605 × 10^8^ Yuan RMB, 2618 × 10^8^ Yuan RMB and 2612 × 10^8^ Yuan RMB, respectively. (2) According to the proportion of the ESVs, the order of the six types of land cover is: forestland > cropland > grassland >water bodies > unutilized land. (3) According to the proportion of various ecosystem service clusters, the ranking is as follows: Regulation services > Support services > Supply services > Cultural services. (4) The contribution of various ecosystem services to the total ESV is ranked as: Soil formation and protection > Gas regulation > Biodiversity conservation > Water conservation > Climate regulation > Raw materials > Waste treatment > Entertainment and culture > Food production. (5) The added value from other land cover types to forest land or water were positive; the added value from other land cover types to urban and rural, industrial and mining, and residential land were negative. The loss of ESV during the period of 2000−2017 was as high as 1628 × 10^6^ Yuan RMB.

Faced with the problems of ecological security and sustainable development caused by land use and land cover, it is necessary to rationally plan and allocate the land resources required for agriculture, forestry, and urban construction, and improve the production efficiency of each industry to achieve high-quality and balanced development [59,60,61]. Based on this research, the following policies were proposed: (1) the government should comprehensively investigate the status quo of cropland landscape as soon as possible and take effective measures to adjust the agricultural industrial structure. Deep processing of economic crops should be promoted to increase the added value of products and improve the supply of ecosystem services. (2) The urban planning layout should be optimized, and the land development process should be strictly controlled to make full use of urban space and improve the regulation services and support services. (3) Classification protection and development strategies should be gradually established to increase the concentration and productivity of forestry industries and increase the total ESV. (4) The water bodies landscape needs to be fully protected and developed to create a complete water bodies landscape system, taking into account the balanced development of agriculture and fisheries.

## Figures and Tables

**Figure 1 ijerph-17-00126-f001:**
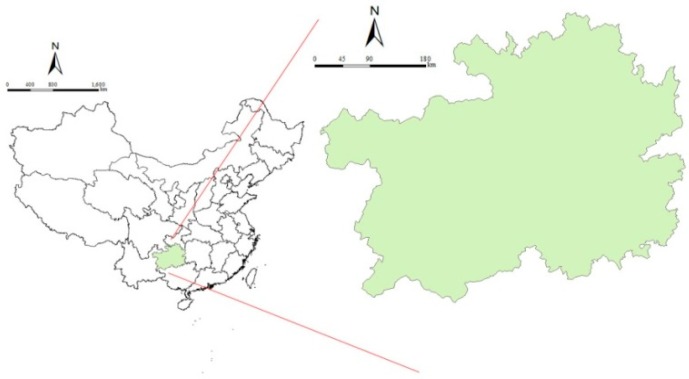
Map of Guizhou Province, China.

**Figure 2 ijerph-17-00126-f002:**
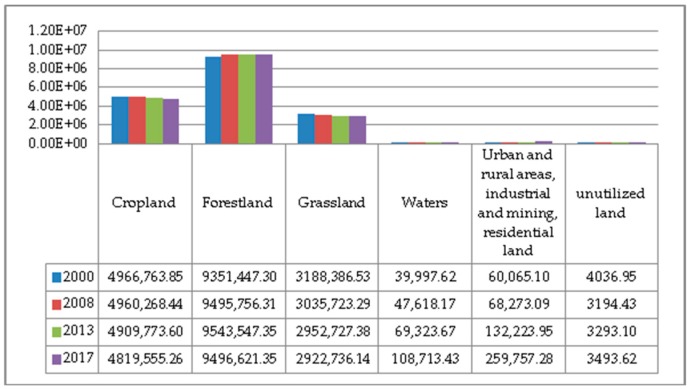
Land use area from 2000 to 2017 (unit: ha).

**Figure 3 ijerph-17-00126-f003:**
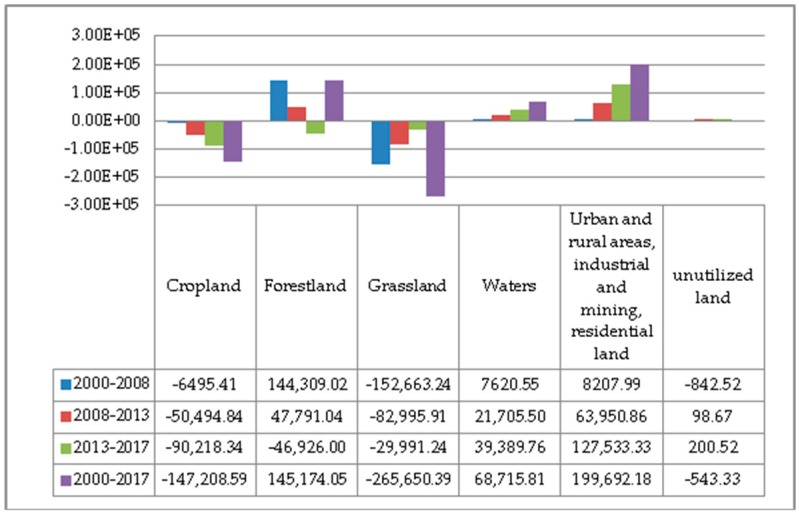
Land use changes from 2000 to 2017 (unit: ha).

**Figure 4 ijerph-17-00126-f004:**
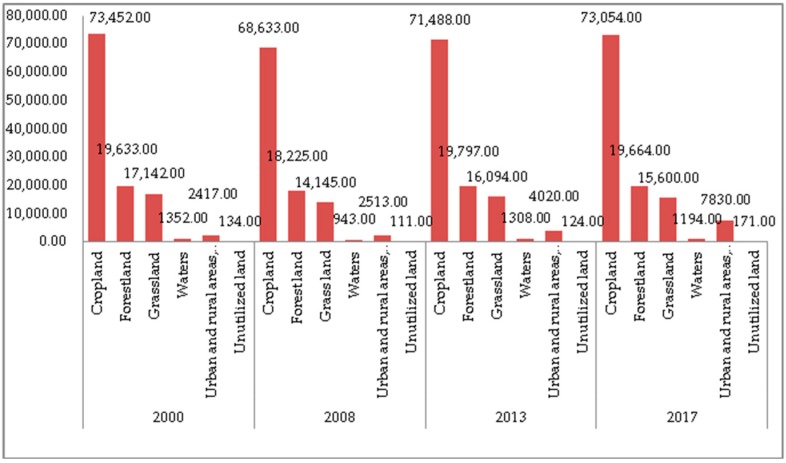
The Patch Number (NP) changes of various types of landscapes.

**Figure 5 ijerph-17-00126-f005:**
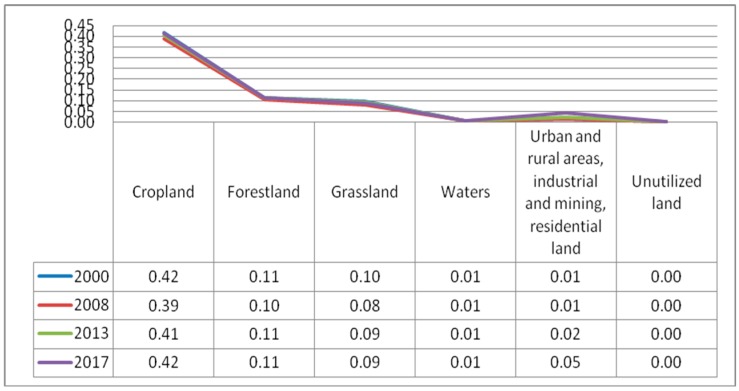
The Patch Density (PD) of various types of landscape.

**Figure 6 ijerph-17-00126-f006:**
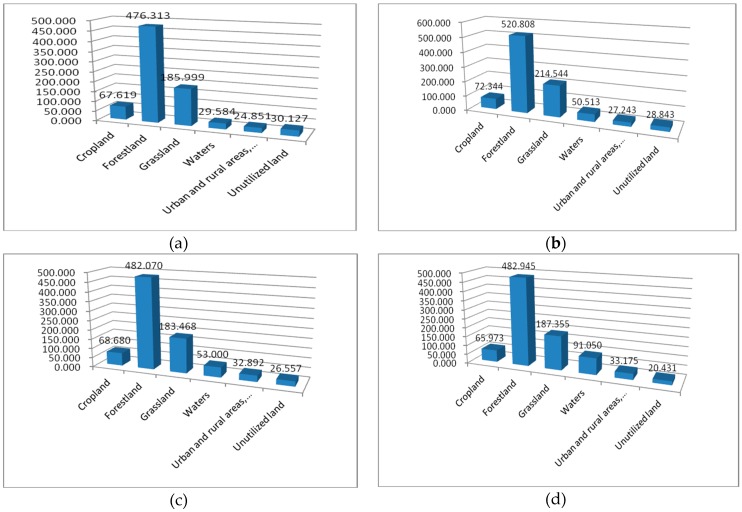
The mean of patch area (AREA-MN) changes of various type of landscapes. (**a**) The AREA-MN in 2000; (**b**) the AREA-MN in 2008; (**c**) the AREA-MN in 2013; (**d**) the AREA-MN in 2017 (unit: ha).

**Figure 7 ijerph-17-00126-f007:**
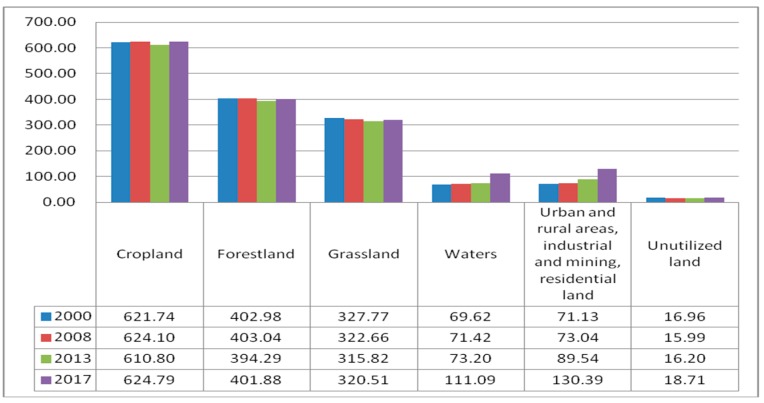
The Landscape Shape Index (LSI) of various types of landscape.

**Figure 8 ijerph-17-00126-f008:**
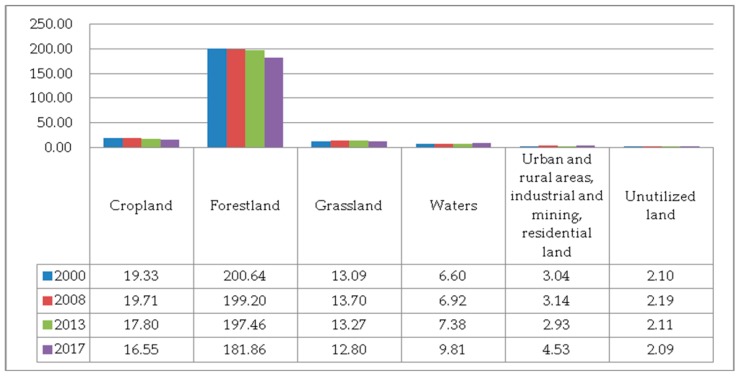
The area-weighted mean (SHAPE_AM) of various types of landscape.

**Figure 9 ijerph-17-00126-f009:**
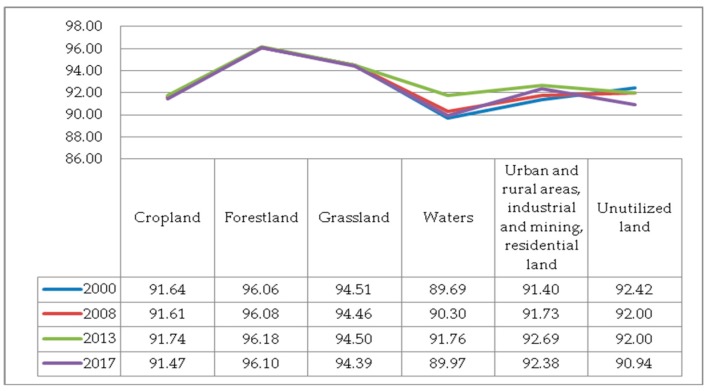
The Aggregation Index (AI) of various types of landscape.

**Figure 10 ijerph-17-00126-f010:**
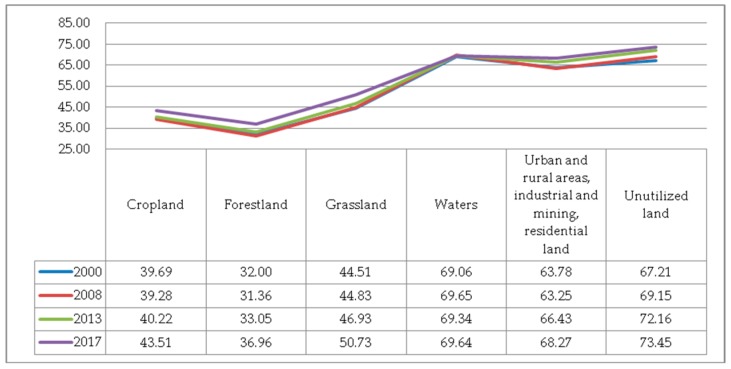
The Interspersion and Juxtaposition index (IJI) of various types of landscape.

**Figure 11 ijerph-17-00126-f011:**
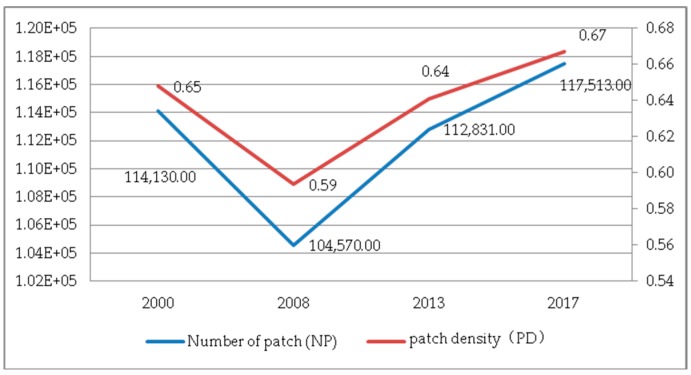
The NP and PD at the landscape level.

**Figure 12 ijerph-17-00126-f012:**
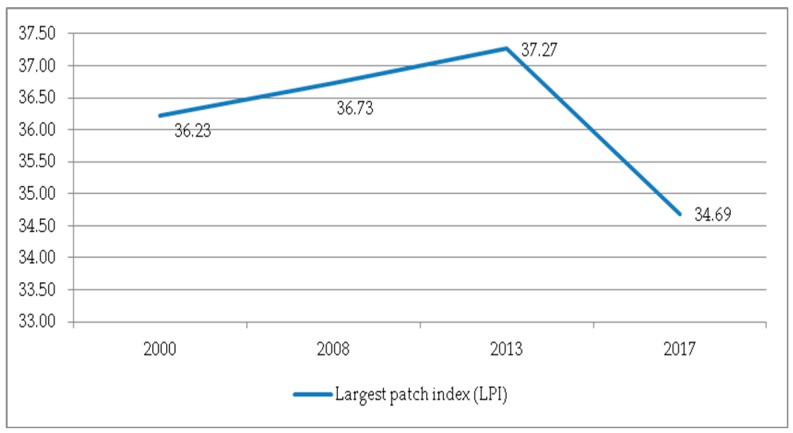
The LPI at the landscape level.

**Figure 13 ijerph-17-00126-f013:**
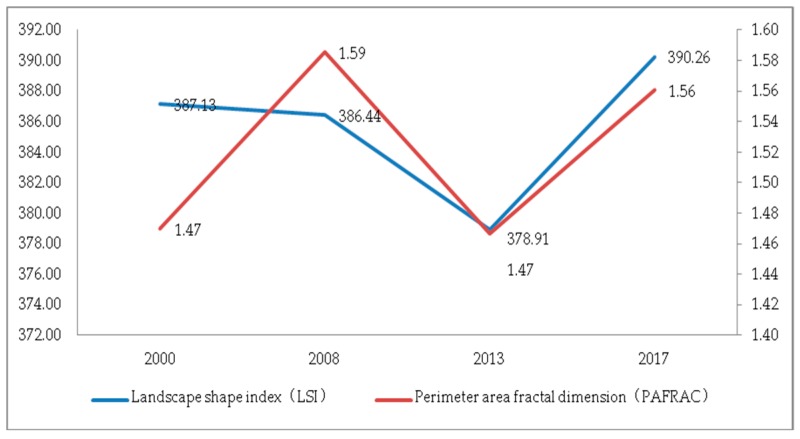
The LSI and perimeter-area fractal dimension (PAFRAC) at the landscape level.

**Figure 14 ijerph-17-00126-f014:**
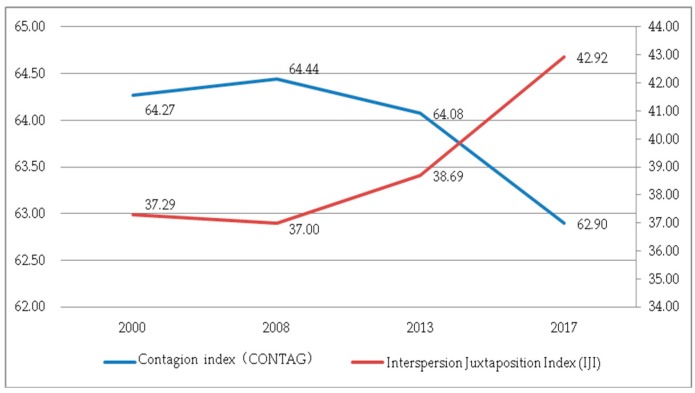
The Contagion Index (CONTAG) and IJI at the landscape level.

**Figure 15 ijerph-17-00126-f015:**
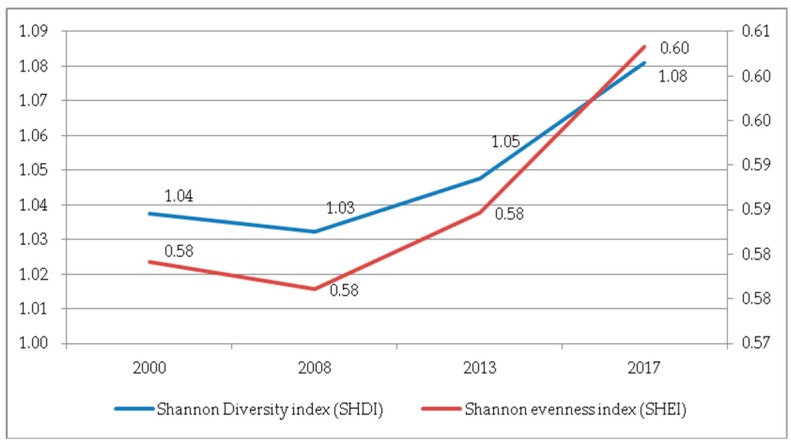
Shannon Diversity Index (SHDI) and Shannon Evenness Index (SHEI) at the landscape level.

**Figure 16 ijerph-17-00126-f016:**
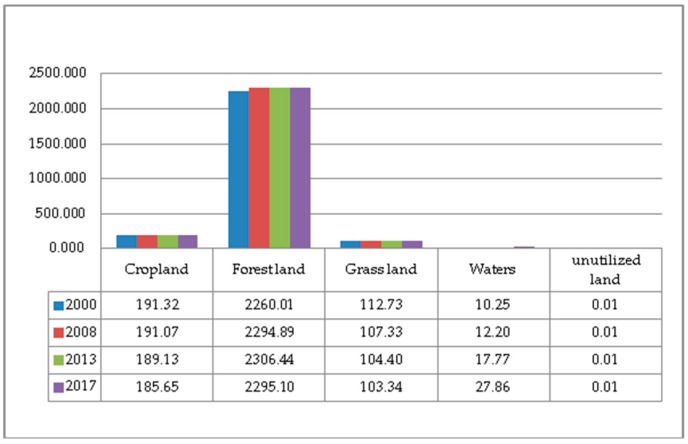
The ESV of various land cover types.

**Figure 17 ijerph-17-00126-f017:**
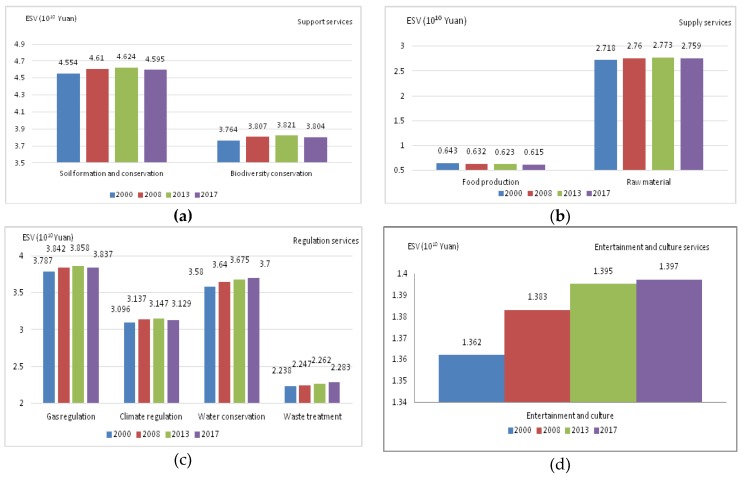
The ESV of various Ecosystem Services. (**a**) The ESV of support services; (**b**) the ESV of supply services; (**c**) the ESV of the regulation services; (**d**) the ESV of entertainment and cultural services.

**Table 1 ijerph-17-00126-t001:** Ecosystem service values (ESVs) of the terrestrial ecosystem (unit: Yuan RMB ha^−1^·a^−1^).

Ecosystem Service Type	Land Use Type				
	Cropland	Forestland	Grassland	Water Bodies	Unutilized Land
Gas regulation	278.71	3871.25	90.34	0.00	0.00
Climate regulation	496.13	2986.38	176.31	256.41	0.00
Water conservation	334.47	3539.38	184.41	11,360.92	16.70
Soil formation and protection	813.90	4313.63	363.83	5.54	11.15
Waste treatment	914.26	1449.00	1217.97	10,134.56	5.54
Biodiversity conservation	395.77	3605.75	594.43	1388.08	189.50
Food production	557.49	110.63	822.98	55.76	5.54
Raw material	55.76	2875.75	4.36	5.54	0.00
Entertainment and culture	5.54	1415.75	80.99	2419.33	5.54
Total	3852.01	24,167.50	3535.62	25,626.13	233.98

**Table 2 ijerph-17-00126-t002:** Gains and losses matrix of ESVs (unit: 10^6^ Yuan RMB).

2000	2017						
	Cropland	Forest land	Grassland	Water bodies	Urban and rural, industrial and mining, and residential land	Unutilized land	Total ESV
Cropland	/	5273.077	−25.645	511.286	−480.742	−0.619	5277.357
Forestland	−4452.025	/	−1441.382	47.283	−1009.41	−9.619	−6865.153
Grassland	37.757	5075.562	/	332.465	−132.785	−0.812	5312.187
Water bodies	−19.075	−1.601	−13.586	/	−5.679	−0.037	−39.978
Urban and rural, industrial and mining, and residential land	8.956	28.185	3.007	12.225	/	0	52.373
Unutilized land	0.573	18.97	0.781	0.518	−0.036	/	20.806
Total ESV	−4423.814	10,394.193	−1476.825	903.777	−1628.652	−11.087	3757.592

**Table 3 ijerph-17-00126-t003:** Sensitivity analysis.

Value Factor Vk	2000		2008		2013		2017	
	ESV Percentage	CS	ESV Percentage	CS	ESV Percentage	CS	ESV Percentage	CS
Cropland Vk ± 50%	3.716	0.074	3.667	0.073	3.612	0.072	3.554	0.071
Forestland Vk ± 50%	43.895	0.878	44.039	0.881	44.054	0.881	43.935	0.879
GrasslandVk ± 50%	2.189	0.044	2.060	0.041	1.994	0.040	1.978	0.040
Water bodies Vk ± 50%	0.199	0.004	0.234	0.005	0.339	0.007	0.533	0.011
Unutilized land Vk ± 50%	0.000	0	0.000	0	0.000	0	0.000	0

**Table 4 ijerph-17-00126-t004:** Correlations of landscape patterns and ESV.

Landscape Index	NP	LSI	AREA_MN	SHAPE_AM	IJI	AI
Correlation coefficient	0.028	0.329	0.948 **	0.999 **	−0.671 **	0.769 **

** Significantly correlated at the 0.01 level

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
