# Peer review of "Quantifying Land Use/Land Cover and Landscape Pattern Changes and Impacts on Ecosystem Services"

_ijerph, 2019, doi:10.3390/ijerph17010126_

Round 1

Reviewer 1 Report

The study focus on the impacts of land use change on ecosystem services. But the estimated methods for ES are simple, which only depends on the experience parameters and land area size. The English writing also need moderately revised. 

Author Response

Response to Reviewer 1 Comments

Dear Mr / Ms,

Thank you very much for your comments and suggestions. Your opinions and suggestions are very insightful and constructive, and are of great value to the improvement of the manuscript. Based on your comments, the author reviewed and revised the manuscript and marked it in the manuscript.  A point-by-point response to the comments is as follow.

Point 1: The study focus on the impacts of land use change on ecosystem services. But the estimated methods for ES are simple, which only depends on the experience parameters and land area size.

Response: Thank you very much for your comments and suggestions. Changes in ecosystem services are usually based on changes in physical and value quantities. In this study, we focused on the changes in the value of ecosystem services and calculated them. For the calculation of the value of ecosystem services, on a large scale regional spatial scale, the value equivalent method of Constanza is a commonly used method. In this study, the value-equivalent method with Chinese regional characteristics was used. Of course, the product-based market price method and the ecosystem service payment willingness method are also optional methods. In the revised manuscript, references have been appropriately added, as well as a description of the assessment methods of ecosystem services, so that the readers can understand the details more clearly. The authors also hope to find and adopt more accurate methods in future research.

Point 2: The English writing also need moderately revised. 

Response: Thank you so much for your suggestion. For English writing problems, the authors have revised and marked several times. For details, please see the manuscript.

Thank you again for your valuable comments, and your comments contributed greatly to the improvement of the manuscript.

Reviewer 2 Report

Dear authors,
With all my respect to your work, these are my comments. Please I encourage you to improve the work. This research work, although interesting, needs to incorporate a series of recommendations so that it can be published.

The introduction is no clear storyline: too many topics wanted to be cover in a no-systematic order. Methodology clearness in achieving the aim of the study is lacking. The results of this study are presented but not deeply and critically discussed.

The theme of sustainability is tangential in this paper, it would be good if the authors reinforced it in some way to connect the study with the objectives of this journal. It would also be necessary to introduce some citations to papers published in this journal, in order to improve this connection.

The description of the methods applied need to be more open, instead of stating that an appropriate process was followed; the process needs to be reported so that the readers may consider its appropriateness themselves. 

The discussion and conclusions do not provide enough detail. Also, the discussion section could benefit from an engagement with the literature that addresses the novelty of the approach, the significance of integrating models and how it might contribute to broader knowledge in the field.

While I suggest these relatively minor revisions of the article, I remain excited about the manuscript’s contribution to IJERPH. I hope the authors receive this review and strengthen the manuscript, and I look forward to seeing it in publication.

Author Response

Response to Reviewer 2 Comments

Dear Mr / Ms,

Thank you very much for your comments and suggestions. Your opinions and suggestions are very insightful and constructive, and are of great value to the improvement of the manuscript. Based on your comments, the author reviewed and revised the manuscript and marked it in the manuscript.  A point-by-point response to the comments is as follow.

Point 1: With all my respect to your work, these are my comments. Please I encourage you to improve the work. This research work, although interesting, needs to incorporate a series of recommendations so that it can be published.

Response: Thank you very much for your comments. Based on your comments, the authors have improved the manuscript. For specific amendments, please see the manuscript and the answers below.

Point 2: The introduction is no clear storyline: too many topics wanted to be cover in a no-systematic order. Methodology clearness in achieving the aim of the study is lacking. The results of this study are presented but not deeply and critically discussed.

Response: Thank you very much for your comments. Based on your comments, the author has organized the introduction to make the subject more visible and more systematic. In the research method section, a description of the research methods was appropriately added to more clearly relate to the research objectives. In the results section, a description of the research results was added and a more in-depth and rigorous discussion was held. For details, please refer to the introduction, research methods and research results of the manuscript.

Point 3: The theme of sustainability is tangential in this paper, it would be good if the authors reinforced it in some way to connect the study with the objectives of this journal. It would also be necessary to introduce some citations to papers published in this journal, in order to improve this connection.

Response: Thank you very much for your comments. Based on your comments, the authors have followed papers published in this journal on the subject. In order to strengthen the theme of sustainability and enhance its relevance to this journal, journal-related research papers have been cited several times in the manuscript.

Point 4: The description of the methods applied need to be more open, instead of stating that an appropriate process was followed; the process needs to be reported so that the readers may consider its appropriateness themselves. 

Response: Thank you very much for your comments. According to your opinion, the author has sorted out the methods used in the paper, and appropriately added descriptions and analysis of various methods so that readers can more clearly understand and judge the applicability and rationality of the method. For specific revisions, see the Materials and Methods section of the manuscript.

Point 5: The discussion and conclusions do not provide enough detail. Also, the discussion section could benefit from an engagement with the literature that addresses the novelty of the approach, the significance of integrating models and how it might contribute to broader knowledge in the field.

Response: Thank you very much for your comments and suggestions. According to your comments, detailed descriptions have been added to the discussion and conclusion sections, including quantitative results and analysis. In addition, relevant discussions and analysis of the method have been added. Please refer to the conclusion and discussion section for specific amendments.

Point 6: While I suggest these relatively minor revisions of the article, I remain excited about the manuscript’s contribution to IJERPH. I hope the authors receive this review and strengthen the manuscript, and I look forward to seeing it in publication.

Response: Thank you very much for your comments and suggestions. Based on your comments, the manuscript has been revised.

Thank you again for your valuable comments, and your comments contributed greatly to the improvement of the manuscript.

Reviewer 3 Report

The authors use a unique approach to quantifying ecosystem service changes through time in Southern China.  The research approach is intriguing and could be replicated in many other environments.  After reading the paper, there are some improvements I recommend prior to publication.  The authors should conduct a more detailed literature review and background to provide context for their methodology and the study area, and, I am not convinced of the significance of their results.  While the method application is intriguing, the size of land use change appear extremely small and could (arguably) be a result of either aggregate remote sensing errors, or, be dispersed throughout the region in such a way that there is a negligible total impact.  The authors need to add in some additional analysis demonstrating that the changes they have observed are significant and meaningful.  More comments below.

Overall Comments-

Grammar and spell check.  Multiple inconsistencies in tense and plurality noticed throughout the manuscript

Authors should present analysis demonstrating the statistical significance of the changes in LU/LC they have documented.

Section Comments

Abstract - need to define ESV acronym before use

line 45 - ‘greatly increased’ is vague.  Elaborate

Lines 86-88 - need to clarify primary and secondary landform classification (a table would help with this).  Right now, it reads as though you have greater than 100% of landform type by area.

General comment on Study Area/Background - more information/context would be useful here (e.g., narrative history of land use and ecosystem service change through time, more discussion on the geology, population, soils, industries, etc. 

Literature/Background - the authors don’t discuss other relevant literature from their study region, nor do they discuss how the methods they’ve chosen have been applied in other systems.  Some additional literature review is needed to provide context

Results - the changes to land cover are relatively small (a few hundred square kilometers) and only represent less than 1% of the total land cover of those types.  What is the significance of these changes on a regional scale?

Figure 6 - Chinese characters in part d?  Are those supposed to be there?

How were the ESV calculations made/assigned?  More detail in the methods for this would be helpful.

Author Response

Response to Reviewer 3 Comments

Dear Mr / Ms,

Thank you very much for your comments and suggestions. Your opinions and suggestions are very insightful and constructive, and are of great value to the improvement of the manuscript. Based on your comments, the author reviewed and revised the manuscript and marked it in the manuscript.  A point-by-point response to the comments is as follow.

Point 1: The authors use a unique approach to quantifying ecosystem service changes through time in Southern China.  The research approach is intriguing and could be replicated in many other environments. 

Response: Thank you very much for your comments, which gave us confidence and motivation.

Point 2: After reading the paper, there are some improvements I recommend prior to publication.  The authors should conduct a more detailed literature review and background to provide context for their methodology and the study area, and, I am not convinced of the significance of their results.  While the method application is intriguing, the size of land use change appear extremely small and could (arguably) be a result of either aggregate remote sensing errors, or, be dispersed throughout the region in such a way that there is a negligible total impact.  The authors need to add in some additional analysis demonstrating that the changes they have observed are significant and meaningful.  More comments below.

Response: Thank you very much for your comments and suggestions. According to your comments, the literature review and research background have been rearranged in the introduction, and relevant content and relevant literature have been appropriately added. In the results section, a detailed analysis of land use changes and transfers has been added to quantify the magnitude and practical significance of data changes. Please refer to the manuscript for specific amendments.

Overall Comments-

Point 3: Grammar and spell check.  Multiple inconsistencies in tense and plurality noticed throughout the manuscript

Response: Thank you very much for your comments. Based on your comments, the authors have thoroughly checked the manuscript and corrected related grammatical issues.

Point 4: Authors should present analysis demonstrating the statistical significance of the changes in LU/LC they have documented.

 Response: Thank you very much for your comments. In the results section, a detailed statistical analysis of the changes in LU / LC is performed. For details, see 3.1 Changes of land use and land cover.

Section Comments

Point 5: Abstract - need to define ESV acronym before use

Response: Thank you very much for your comments. The revision has been completed based on your comments.

Point 6: line 45 - ‘greatly increased’ is vague.  Elaborate

Response: Thank you very much for your comments. The revision has been completed based on your comments.

Point 7: Lines 86-88 - need to clarify primary and secondary landform classification (a table would help with this).  Right now, it reads as though you have greater than 100% of landform type by area.

Response: Thank you very much for your comments. The description of landform classification has been revised and supplemented. For details, see the study area of the manuscript.

Point 8: General comment on Study Area/Background - more information/context would be useful here (e.g., narrative history of land use and ecosystem service change through time, more discussion on the geology, population, soils, industries, etc. 

Response: Thank you very much for your comments. Added description and analysis of changes in land use and ecosystem services, as well as geology, population, soil, and industrial development in the study area. See the Study Area section for details.

Point 9: Literature/Background - the authors don’t discuss other relevant literature from their study region, nor do they discuss how the methods they’ve chosen have been applied in other systems.  Some additional literature review is needed to provide context

Response: Thank you very much for your comments. In the study area, some literatures related to the research area were added; in the methods section, literatures related to methods were added, and moderate discussions were conducted. See the manuscript for details.

Point 10: Results - the changes to land cover are relatively small (a few hundred square kilometers) and only represent less than 1% of the total land cover of those types.  What is the significance of these changes on a regional scale?

Response: Thank you very much for your comments. According to the results of the study, the final change ratio of various land cover types is relatively small, generally not exceeding 1%. However, this result is the final result after various land cover types are transferred in and out, and the proportion of transfer in and transfer out is relatively large. In order to more accurately express the details of land cover changes, a quantitative discussion and analysis of the transfer of various land cover types in and out was added to the results section of the manuscript. See the manuscript for specific revisions.

Point 11: Figure 6 - Chinese characters in part d?  Are those supposed to be there?

Response: Thank you very much for your comments. This is a mistake in writing the manuscript. Chinese characters in the revised draft have been removed.

Point 12: How were the ESV calculations made/assigned?  More detail in the methods for this would be helpful.

Response: Thank you very much for your comments and suggestions. Changes in ecosystem services are usually based on changes in physical and value quantities. In this study, we mainly focused on the changes in value and calculated them. For the calculation of the value of ecosystem services, on a large-scale regional spatial scale, the value-equivalent method based on Constanza and other pioneering methods is a commonly used method. In this study, the value-equivalent method improved based on China's regional characteristics was used. This is discussed in the manuscript. Of course, the product-based market price method and the ecosystem service payment willingness method are also optional methods. In the revised manuscript, references have been appropriately added, as well as a description of the assessment methods of ecosystem services, so that the reader can understand the details more clearly. Of course, the authors hope to find and adopt more accurate methods in future research.

Thank you again for your valuable comments, and your comments contributed greatly to the improvement of the manuscript.

Round 2

Reviewer 3 Report

The authors addressed my concerns regarding issue with the text of the manuscript.  I don't know if it is the format of the figures or an issue with the program I am reading the manuscript in (Adobe Acrobat Reader) but none of the graphs have y-axes or y axes labels.  This obviously needs to be addressed prior to publication.